# Estimating dispensable content in the human interactome

Mohamed Ghadie [1] & Yu Xia[1]

Protein-protein interaction (PPI) networks (interactome networks) have successfully advanced our knowledge of molecular function, disease and evolution. While much progress has been made in quantifying errors and biases in experimental PPI datasets, it remains unknown what fraction of the error-free PPIs in the cell are completely dispensable, i.e., effectively neutral upon disruption. Here, we estimate dispensable content in the human interactome by calculating the fractions of PPIs disrupted by neutral and non-neutral mutations. Starting with the human reference interactome determined by experiments, we construct a human structural interactome by building homology-based three-dimensional structural models for PPIs. Next, we map common mutations from healthy individuals as well as Mendelian disease-causing mutations onto the human structural interactome, and perform structure-based calculations of how these mutations perturb the interactome. Using our predicted as well as experimentally-determined interactome perturbation patterns by common and disease mutations, we estimate that <~20% of the human interactome is completely dispensable.

[1] Department of Bioengineering, McGill University, Montreal, QC, Canada. Correspondence and requests for materials should be addressed to Y.X. (email: brandon.xia@mcgill.ca)

Protein–protein interactions (PPIs) are a central type of molecular interactions in the cell which collectively form the interactome network. Significant progress has been made toward mapping interactome networks for several species including human[1,2]. These networks have been highly successful in providing insights into molecular function[3,4], disease[1,5–8], and evolution[9–13]. While much work has been done in quantifying errors and biases in experimental PPI datasets[14–16], it remains unknown what fraction of the error-free PPIs in the cell are completely dispensable, i.e., effectively neutral upon disruption. Unlike erroneous PPIs which are experimental false positives, completely dispensable PPIs are true physical interactions which may or may not be associated with well-defined molecular functions. Furthermore, we draw a clear distinction between completely dispensable PPIs and non-specific PPIs. Non-specific PPIs have been used in the literature to describe non-stereospecific interactions or transient interactions that may well be crucial to cellular function[17,18], interactions that involve promiscuous binding of a protein to many partners[19], or interactions that may have no function[20]. While the last definition of non-specific PPIs comes close to our definition of completely dispensable PPIs, the first two definitions of non-specific PPIs are very different from our definition of completely dispensable PPIs. The unique defining feature of completely dispensable PPIs is that they do not measurably affect organismal fitness upon disruption.

The question of dispensable content in interactome networks is of utmost importance to cell systems biology, with widely diverging opinions in the literature. On the one hand, current systems biology studies using interactome networks to understand human disease depend crucially on the assumption that completely dispensable PPIs do not dominate the human interactome[1,21,22]. On the other hand, the existence of completely dispensable PPIs is well-anticipated by molecular evolution and population genetics theory[23,24], as well as strongly supported by empirical analysis of genome-wide data[25–27]. PPIs that are completely dispensable are introduced into and maintained in the interactome by non-adaptive processes, when purifying selection is not strong enough to maintain the perfect interactome in the presence of mutation and genetic drift, especially in species with small population sizes[23,24,27]. Completely dispensable PPIs can lead to increasing robustness of the interactome network against mutations in that their elimination by mutations does not induce any measurable change in organismal fitness[8,13,28,29]. This type of interactome robustness against mutations, which is unique to completely dispensable PPIs, is distinct from another type of interactome robustness against mutations, where PPIs in the interactome network are preserved in the presence of mutations at the binding interface[11,13,30]. Because completely dispensable PPIs are effectively neutral upon disruption, they tend to evolve more quickly than other PPIs[25]. Given the estimate that much of the human genome may be "junk" DNA under neutral evolution[31,32], it is possible that a large fraction of the human interactome may be "junk" interactions as well[16,23–25].

Here, in an effort to resolve the long-standing debate over completely dispensable contents in interactome networks, we estimate the overall fraction of PPIs in the human interactome that are effectively neutral upon disruption by mutation. Starting with a high-quality, experimentally determined human reference interactome, we construct a human structural interactome by building three-dimensional (3D) structural models for known human PPIs and annotating PPI interfaces at the residue level using template-based homology modeling. Similar structural network biology approaches have been previously used to provide insights into protein function, disease, and evolution[33–41]. Next, we map common mutations from healthy individuals as well as Mendelian disease-causing mutations onto the human

structural interactome, and perform structure-based prediction of the edgotype[42] for each mutation, i.e., the precise pattern of interactome perturbation as the result of each mutation. We integrate these results to calculate the probabilities for common mutations (assumed to be neutral) and disease-causing mutations (assumed to be mildly deleterious) to disrupt human PPIs, and then apply Bayes' theorem to calculate the probabilities for human PPIs to be neutral or non-neutral upon disruption[13]. Our calculations reveal that overall <~20% of the human interactome is completely dispensable, i.e., effectively neutral upon disruption. Finally, instead of using computationally predicted edgotypes for mutations, we repeat our calculations using experimentally determined edgotypes for mutations[8]. Our dispensable PPI estimate remains broadly consistent despite minimal overlap in protein space covered by computational and experimental edgotyping data.

## Results

**Construction of the human structural interactome**. We started with two high-quality, experimentally determined human reference interactomes: the HI-II-14 interactome[43] consisting of PPIs identified in yeast two-hybrid (Y2H) screens, and the IntAct interactome consisting of PPIs reported in the IntAct database[44] by at least two independent experiments in the literature. From each of the two human reference interactomes, we constructed a human structural interactome by building 3D structural models for known PPIs via homology modeling, using experimentally determined PPI structural templates in the Protein Data Bank (PDB)[45] (Fig. 1). As a result, we obtained two high-resolution human structural interactomes: the HI-II-14 structural interactome (Y2H-SI) consisting of 486 PPIs among 573 proteins with their binding interfaces resolved at the residue level (Supplementary Data 1a and 2a), and the IntAct structural interactome (IntAct-SI) consisting of 3333 PPIs among 2654 proteins with their binding interfaces resolved at the residue level (Supplementary Data 1b and 2b). The high quality of our structurally annotated interactomes is confirmed by high functional similarity and tissue co-expression among interacting proteins (Supplementary Fig. 1a–d). Our structural interactomes are modeled from a diverse set of PDB structures (Supplementary Fig. 1e), with >40 residues on average mapped to an interface per protein (Supplementary Fig. 1f).

**Geometry-based prediction of mutation edgotypes**. We mapped 3705 Mendelian disease-causing missense mutations from ClinVar[46] and 28,788 common missense mutations not associated with disease from dbSNP[47] onto our two structural interactomes: Y2H-SI and IntAct-SI (Fig. 1). Overall, Y2H-SI carries 145 disease mutations and 376 non-disease mutations (Supplementary Data 3a, b), and IntAct-SI carries 908 disease mutations and 2394 non-disease mutations (Supplementary Data 3c, d). These mutations span a significant part of the human structural interactome, covering ~32% of proteins in Y2H-SI and ~41% of proteins in IntAct-SI.

Next, we used the structural interactome to perform geometry-based prediction of the edgotype for each mutation, i.e., the precise pattern of interactome perturbation as the result of each mutation. Mutations can be either edgetic (i.e., disrupt specific PPIs by disrupting binding interfaces), quasi-null (i.e., disrupting all PPIs by disrupting overall protein stability), or quasi-wildtype (i.e., do not disrupt any PPIs)[8]. We predict that a mutation edgetically disrupts a PPI if and only if the mutation occurs on the interface mediating that PPI (Figs. 1 and 2a). In Y2H-SI, we predicted 5.1% (19 out of 376) of non-disease mutations to be edgetic and 18.6% (27 out of 145) of disease mutations to be

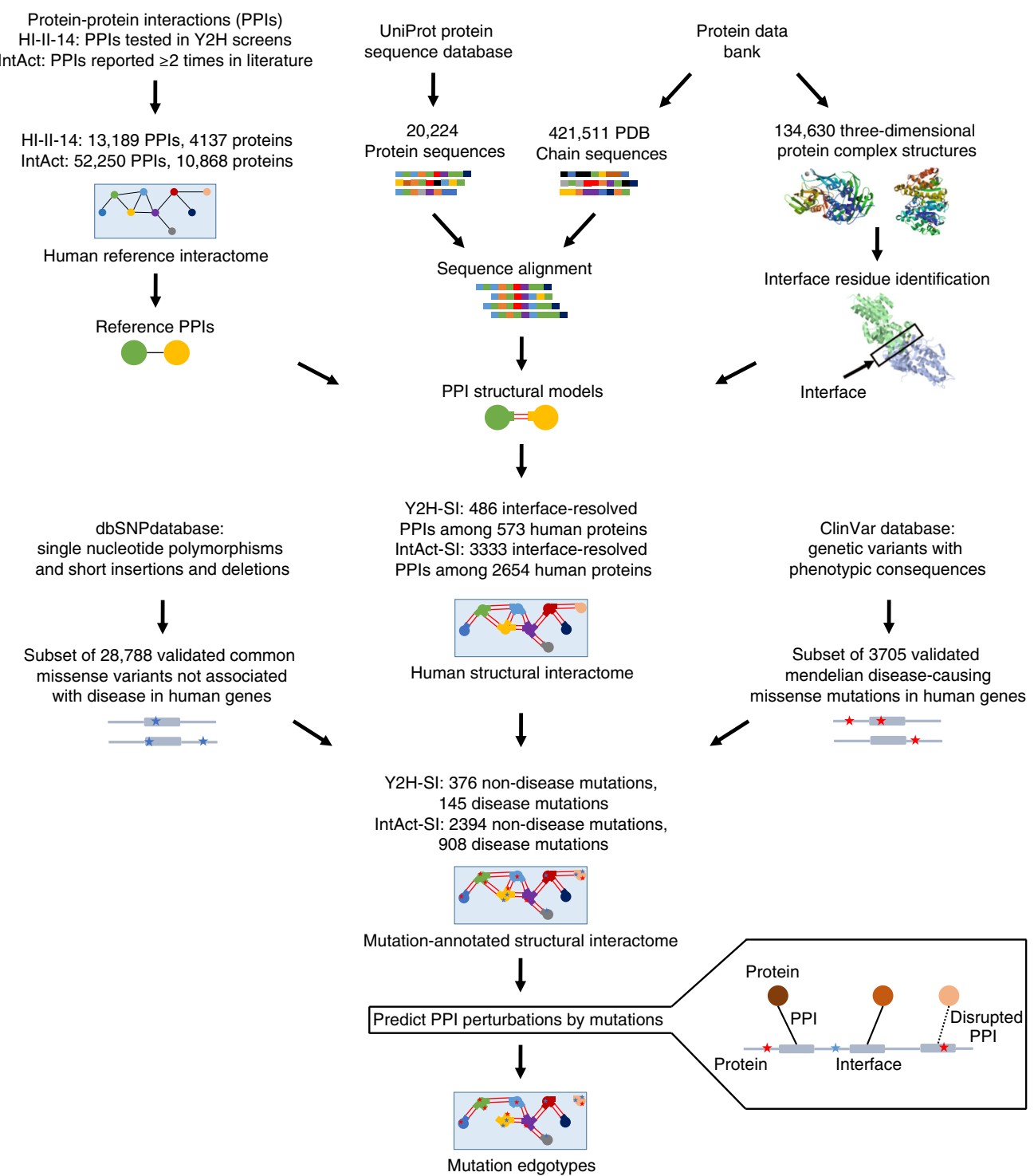

**Fig. 1** Pipeline for the computational prediction of mutation edgotypes. Computational pipeline used for construction of the human structural interactomes (Y2H-SI and IntAct-SI) from the human reference interactomes (HI-II-14 and IntAct) and subsequent prediction of mutation-induced interactome perturbations and mutation edgotypes

edgetic (Fig. 2b; Supplementary Data 3a, b). In IntAct-SI, we predicted 6.9% (164 out of 2394) of non-disease mutations to be edgetic and 15.4% (140 out of 908) of disease mutations to be edgetic (Fig. 2b; Supplementary Data 3c, d). In comparison, in the experimental study of Sahni et al.[8], it was found that 4.3% (2 out of 47) of non-disease mutations are edgetic and 31.5% (62 out of 197) of disease mutations are edgetic (Fig. 2b). Thus, our computational results are consistent with experimental results in

that disease mutations are significantly more likely to be edgetic than non-disease mutations ($p < 10^{-4}$ for all cases, two-sided Fisher's exact test).

**Geometry-based calculation of dispensable PPI content**. We used the mutation edgotypes predicted above to estimate the fraction of PPIs in the human interactome that are completely dispensable, i.e, effectively neutral upon disruption, following the

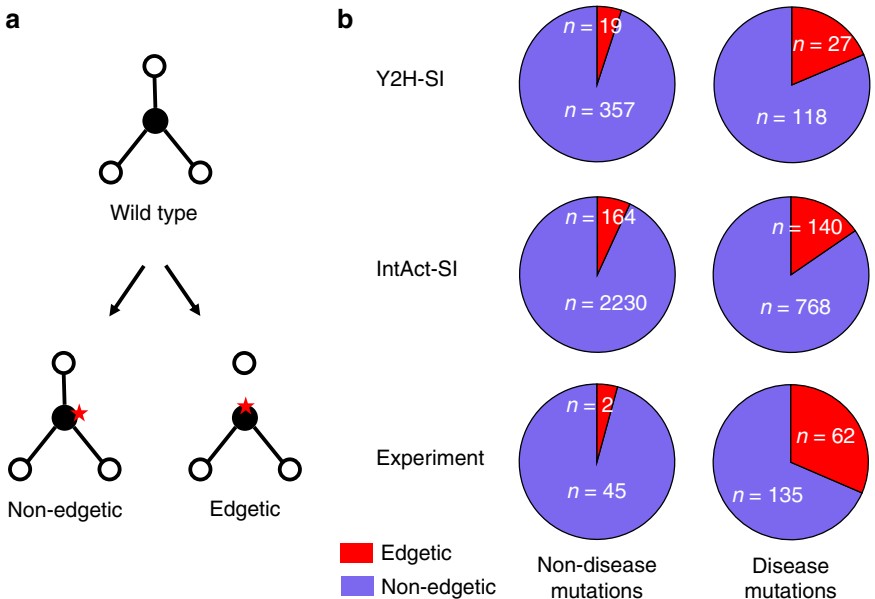

**Fig. 2** Mutation edgotypes determined by geometry-based predictions and experiments. **a** Graphical illustration of edgetic and non-edgetic mutations. **b** Fraction of edgetic mutations among common non-disease mutations (left) and among Mendelian disease-causing mutations (right), determined by geometry-based prediction of mutation edgotypes in the two human structural interactomes, Y2H-SI and IntAct-SI, and by experiments[8]. Red slices represent edgetic mutations, and purple slices represent non-edgetic mutations. Source data are provided as a Supplementary Data file

procedure we had previously developed[13]. We assume that mutations are either effectively neutral (similar to synonymous mutations), mildly deleterious, or strongly detrimental (similar to nonsense mutations that introduce premature stop codons). In addition, we assume that common mutations from healthy individuals are effectively neutral, that Mendelian disease-causing mutations are mildly deleterious on average, and that strongly detrimental mutations are quasi-null (i.e., disrupt overall protein stability) rather than edgetic.

Using our predicted mutation edgotypes in Y2H-SI from the previous section, we obtained the probabilities for effectively neutral (N), mildly deleterious (M), and strongly detrimental (S) mutations to be edgetic (E): $P(E|N) = 5.1\%$, $P(E|M) = 18.6\%$, $P(E|S) = 0$ (Fig. 2b). Furthermore, we obtained from Kryukov et al.[48] the probabilities for new missense mutations to be effectively neutral (N), mildly deleterious (M), or strongly detrimental (S): $P(N) = 27\%$, $P(M) = 53\%$, $P(S) = 20\%$. We then integrated these numbers to calculate the probability for new missense mutations to be edgetic (E): $E) = P(E|N)P(N)+P(E|M)P(M) + P(E|S)P(S) = 11.2\%$. Finally, using Bayes' theorem $P(A|B) = P(B|A)P(A)/P(B)$, we calculated the probability for edgetic mutations (E) to be effectively neutral (N): $P(N|E) = P(E|N)P(N)/P(E) = 12.1\%$. Thus, given that most (54%) edgetic mutations disrupt one PPI in Y2H-SI, we estimated that ~12.1% of the human interactome is completely dispensable, i.e., effectively neutral upon disruption, with a 95% confidence interval of 7.4–19.4% (Fig. 3).

Next, we repeated the same calculation using our predicted mutation edgotypes in IntAct-SI from the previous section (Fig. 2b), and estimated that ~18.5% of the human interactome is completely dispensable, with a 95% confidence interval of 15.5–21.9% (Fig. 3). Finally, we repeated the same calculation using the experimental mutation edgotype data from Sahni et al.[8] (Fig. 2b), and estimated that ~6.4% of the human interactome is completely dispensable, with a 95% confidence interval of 1.7–21.4% (Fig. 3). These three dispensable PPI content estimates obtained from predicted and experimental mutation edgotypes are broadly consistent with each other.

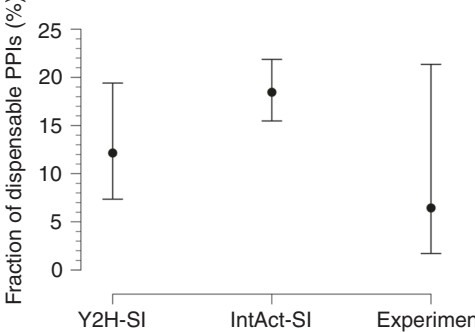

**Fig. 3** Dispensable PPI content estimated from geometry-based predictions and experiments. Fraction of completely dispensable PPIs in the human interactome, estimated from mutation edgotypes obtained by geometry-based predictions in the two human structural interactomes (Y2H-SI and IntAct-SI), and from mutation edgotypes obtained by experiments[8]. Bars represent 95% confidence intervals

**Physics-based calculation of dispensable PPI content**. Our geometry-based mutation edgotype predictions described above assume that both mildly deleterious disease mutations (M) and effectively neutral non-disease mutations (N) located at a PPI interface disrupt that PPI with the same probability $\gamma_M = \gamma_N = 100\%$. This assumption is inaccurate, because disease mutations and non-disease mutations impact PPI stability differently due to their different physicochemical properties on average. Indeed, when we calculated the substitution scores for all 28,788 non-disease missense mutations and 3705 disease missense mutations in human using the PAM30 substitution matrix, we found that disease mutations tend to have a lower substitution score than non-disease mutations ($p < 10^{-6}$, two-sided bootstrap test with 1,000,000 resamplings; Fig. 4a), indicating that disease mutations tend to be more radical than non-disease mutations.

Hence, we performed physics-based calculation of $\gamma_M$ and $\gamma_N$, the probabilities for disease and non-disease interfacial mutations to disrupt the corresponding PPI. We first focused on Y2H-SI.

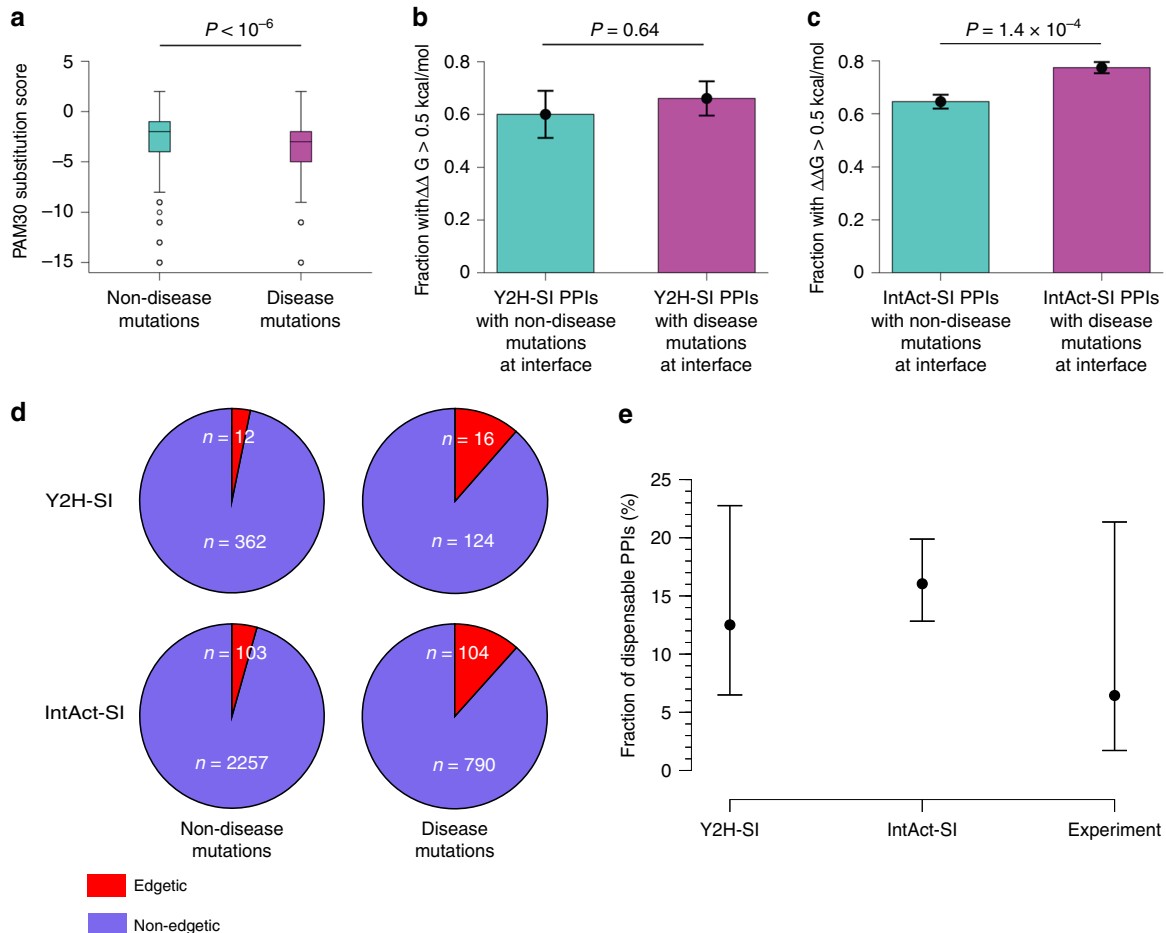

**Fig. 4** Dispensable PPI content estimated from physics-based predictions. **a** Substitution score calculated using the PAM30 substitution matrix for all non-disease and disease-causing missense mutations in human. Center line represents the median value. Lower and upper bounds of the box represent the lower and upper quartile values, respectively. Whiskers represent the data range. Statistical significance was calculated using a two-sided bootstrap test with 1,000,000 resamplings. **b**, **c** Fraction of PPIs with mutations at the interface in the two human structural interactomes, **b** Y2H-SI and **c** IntAct-SI, that have a change in binding free energy ($\Delta\Delta G$) larger than 0.5 kcal mol$^{-1}$ upon mutation. $\Delta\Delta G$ values were calculated from PPI structural models using BindProfX[49]. Error bars represent standard errors of the fraction. Statistical significance was calculated using a two-sided Fisher's exact test. Source data are provided as a Supplementary Data file. **d** Fraction of edgetic mutations among common non-disease mutations (left) and among Mendelian disease-causing mutations (right), determined by physics-based prediction of mutation edgotypes in the two human structural interactomes, Y2H-SI and IntAct-SI. Red slices represent edgetic mutations, and purple slices represent non-edgetic mutations. Source data are provided as a Supplementary Data file. **e** Fraction of completely dispensable PPIs in the human interactome with 95% confidence interval, estimated from physics-based prediction of mutation edgotypes in the two human structural interactomes (Y2H-SI and IntAct-SI), and from mutation edgotypes obtained by experiments[8]. Bars represent 95% confidence intervals

For each interfacial mutation, we calculated the change in binding free energy ($\Delta\Delta G$) caused by that mutation from the PPI structural model using BindProfX[49], which has been shown to accurately reproduce experimental $\Delta\Delta G$ measurements[49]. The PPI is considered disrupted by the mutation if and only if $\Delta\Delta G$ >0.5 kcal mol$^{-1}$. We performed this physics-based calculation on all interfacial mutations to obtain $\gamma_M = 66\%$ and $\gamma_N = 60\%$ (Fig. 4b; Supplementary Data 4a, b). Using these physics-based PPI perturbation predictions, we found 3.2% (12 out of 374) of non-disease mutations to be edgetic and 11.4% (16 out of 140) of disease mutations to be edgetic (Fig. 4d; Supplementary Data 5a, b), and we estimated that ~12.5% of the human interactome is completely dispensable, with a 95% confidence interval of 6.5–22.8% (Fig. 4e).

Next, we repeated the same physics-based calculation on IntAct-SI. We obtained $\gamma_M = 77\%$ and $\gamma_N = 65\%$ (Fig. 4c; Supplementary Data 4c, d). Using these physics-based PPI perturbation predictions, we found 4.4% (103 out of 2360) of

non-disease mutations to be edgetic and 11.6% (104 out of 894) of disease mutations to be edgetic (Fig. 4d; Supplementary Data 5c, d), and we estimated that ~16% of the human interactome is completely dispensable, with a 95% confidence interval of 12.8–19.9% (Fig. 4e). These adjusted mutation edgotype predictions and corresponding dispensable PPI content estimates remain consistent with those obtained from mutation edgotype experiments[8].

## Discussion

Our estimates of dispensable PPI content were derived from PPI perturbation patterns (edgotypes) in diverse human interactome datasets (HI-II-14 and IntAct). These PPI perturbation patterns were obtained by computation as well as by experiment. Our computational predictions complement experimental data as they probe different subsets of the human protein space, with <7% of computational edgotyping data covered by experiments. Despite

such minimal overlap in protein coverage, our dispensable PPI estimates are broadly consistent with one another (~13% from Y2H-SI, ~16% from IntAct-SI, and ~6% from experiment). Indeed, the 95% confidence intervals for all three estimates overlap below ~20%. Furthermore, the dispensable PPI content obtained using physics-based calculations on the combined network of Y2H-SI and IntAct-SI remains below ~20% (15.8% with a 95% confidence interval of 12.6–19.6%). Taking these results together, we conclude that up to ~20% of the human interactome is completely dispensable, i.e., effectively neutral upon disruption.

PPI datasets are known to contain experimental false positives (erroneous PPIs)[14–16]. These include, among others, non-reproducible experimental artifacts, in vitro physical interactions that do not occur in vivo (more likely to occur in Y2H experiments), and pairs of proteins from the same complex that do not directly interact with each other (more likely to occur in affinity capture experiments). Our goal here is to focus on real PPIs that are free from these errors, and estimate the fraction of these error-free PPIs that are effectively neutral upon disruption. We used several methods to minimize such false positive errors. First, we started from experimentally determined PPIs, rather than computationally predicted PPIs. Second, we used the HI-II-14 dataset, which was subjected to multiple Y2H screens and other quality control measures, and is similar in quality to a gold-standard dataset of literature-derived PPIs[8,43]. Third, for the IntAct dataset, we only considered high-quality PPIs reported by at least two independent experiments in the literature. Fourth, we further reduced false positive errors by focusing on those PPIs for which we can build homology models using experimentally determined 3D structural templates of interacting proteins in PDB.

Despite these efforts, it remains a possibility that the false positive rates of our structural interactome datasets are non-negligible. These erroneous PPIs do not physically occur in the cell with detectable phenotypic consequences, and hence they are typically unable to distinguish deleterious mutations from neutral mutations. Consequently, the error-free portion of the PPI dataset must distinguish deleterious mutations from neutral mutations better than the average performance of the entire PPI dataset. Since in our case, higher predictive power of PPIs for deleterious mutations leads to lower estimate of dispensable PPI content, the fraction of error-free PPIs that are completely dispensable will be even lower than calculated from the entire PPI dataset. Thus, our calculated ~20% completely dispensable content in the human interactome represents an upper bound in the presence of errors in PPI datasets.

Our structure-based mutation edgotype computations contain several potential biases and approximations. First, literature-derived PPIs are biased toward interactions with functional and disease importance. We address this bias by additionally examining systematic PPI datasets such as HI-II-14. Second, experimentally determined 3D structures of interacting proteins are biased toward PPIs with functional and disease importance. We partially address this bias by using homology models in addition to experimental 3D structures of PPIs. Third, our mutation edgotype predictions involve numerous approximations. We address this issue by complementing geometry-based calculations with physics-based calculations, and by using the well-known BindProfX[49] method that has been shown to accurately reproduce experimental measurements of binding free energy change upon mutation. In addition to the BindProfX method, we also repeated our physics-based calculations of dispensable PPI content using $\Delta\Delta G$ values calculated by another well-known method FoldX[50] (Supplementary Data 6a–d), which produces high-quality $\Delta\Delta G$ values when benchmarked using the gold-standard dataset of SKEMPI[51] (Pearson correlation coefficient between

predicted versus experimental $\Delta\Delta G$ is 0.50 for co-crystal structures, and 0.42 for homology models). In Y2H-SI, we found that 11.1% of the human interactome is completely dispensable, with a 95% confidence interval of 4.7–24.1%. In IntAct-SI, we found that 13% of the human interactome is completely dispensable, with a 95% confidence interval of 9.5–17.5%. These FoldX-based estimates of dispensable PPI content remain in broad agreement with our BindProfX-based estimates (12.5% in Y2H-SI, and 16% in IntAct-SI). Fourth, we compare our mutation edgotype computations with experiments. The experimental mutation edgotyping data, while not perfect (low coverage, possible false positives, false negatives), are nonetheless not affected by any of the aforementioned biases and approximations present in our predictions. The broad agreement between computation and experiment indicates that our estimates are robust against these biases and approximations.

Our calculations of dispensable PPI content make the reasonable assumption that strongly detrimental mutations are quasi-null rather than edgetic. While it is difficult to calculate the precise probability for strongly detrimental mutations to be edgetic in the absence of genome-wide data, including such probability in our calculations will only further decrease our estimate of dispensable PPI content. This is because the fraction of PPIs effectively neutral upon disruption is inversely proportional to the overall fraction of missense mutations that are edgetic. Hence, including some strongly detrimental mutations as edgetic in our calculations will increase the overall fraction of missense mutations that are edgetic, resulting in a smaller estimate of dispensable PPI content.

Our calculations of dispensable PPI content assume that each edgetic mutation disrupts one PPI, which is true for most mutations in Y2H-SI (61%) and IntAct-SI (63%). We further repeated our calculations using physics-based mutation edgotype predictions in Y2H-SI and IntAct-SI, this time replacing the fractions of mutations that are edgetic for both disease and non-disease mutations by the fractions of mutations that are mono-edgetic, i.e., those that disrupt only one PPI. Applying our modified calculation to Y2H-SI, we estimated that ~14.5% of the human interactome is completely dispensable with a 95% confidence interval of 6.3–30.1%. Applying the same modified calculation to IntAct-SI, we estimated that ~21.3% of the human interactome is completely dispensable with a 95% confidence interval 16.2–27.6%. These estimates remain very close to our previous estimates. A similar calculation on the experimental dataset of Sahni et al.[8] is not possible, as there are only two non-disease mutations in the dataset that are edgetic, both of which disrupt multiple PPIs and none of which are mono-edgetic.

The most accurate way of calculating dispensable PPI content is to measure the fitness change of the cell by systematically deleting PPIs one at a time. In the absence of such experiments, our calculations offer the next best solution by examining phenotypic consequences of edgetic mutations that disrupt as few as one PPI at a time, while maintaining all other aspects of protein biophysics and cell biology (e.g., protein stability, protein expression, and other protein interactions). Our calculations clearly distinguish edgetic mutations from quasi-null mutations, which, by disrupting overall protein stability, cause complex cellular and phenotypic changes beyond those explainable by simple PPI disruptions. Our structure-based predictions offer a clear definition of edgetic mutations, where mutations at interfacial sites are considered edgetic if they disrupt at least one PPI. On the other hand, the definition of edgetic mutations is less straightforward in the experimental dataset of Sahni et al.[8] due to lack of structural information. There, mutations are considered edgetic if they disrupt at least one PPI but not all PPIs associated with the protein, and mutations that disrupt all PPIs are

considered quasi-null. This definition is not completely accurate because some edgetic mutations may disrupt all PPIs by disrupting the binding interface without affecting protein stability, and they will be misclassified as quasi-null mutations. To test the effect of such potential misclassification on our experiment-based dispensable PPI content estimate, we repeated our calculations using the experimental dataset of Sahni et al.[8], this time treating all quasi-null mutations as edgetic mutations. Using this modified calculation, we estimated that ~7% of the human interactome is completely dispensable with a 95% confidence interval of 2.9–16.3%. This estimate remains very close to our previous estimate obtained from experiments.

Our estimate of dispensable content in the human reference interactome is robust to the presence of gain-of-function mutations. Gain-of-function mutations are capable of driving diverse disease phenotypes by creating new molecular interactions[29,52]. A classic example is sickle cell anemia, where a mutation on the surface of the hemoglobin molecule can cause it to bind to other hemoglobin molecules[29]. Many other examples of gain-of-function mutations have been identified as important in cancer[53,54], neurodegenerative diseases[55], as well as other diseases[56]. Such gain-of-function mutations are challenging to detect systematically, either by experiment or by computation. A recent genome-wide screen suggests that gain-of-interaction mutations are ~30 times less likely to occur in human disease than edgetic loss-of-interaction mutations[8]. Our definition of completely dispensable interactions only refers to pre-existing PPIs in the reference interactome which are neutral upon elimination by mutation, and is independent of the extent of gain-of-function mutations. Furthermore, our Bayesian formulation for estimating dispensable PPI content is robust to gain-of-function mutations. The three prior probabilities P(N), P(M), and P(S), for new missense mutations to be neutral (N), mildly deleterious (M), and strongly detrimental (S), are obtained from the literature using procedures that are robust to gain-of-function mutations[48]. In addition, the other three conditional probabilities in our Bayesian framework P(E|N), P(E|M), and P(E|S), for neutral (N), mildly deleterious (M), and strongly detrimental (S) mutations to edgetically eliminate PPIs (E), are also independent of the extent of gain-of-function mutations. While gain-of-function mutations are beyond the scope of our current study and do not affect our estimate of completely dispensable content among pre-existing PPIs in the reference interactome, our Bayesian framework can be extended in the future to the calculation of completely dispensable content in de novo PPIs newly created by gain-of-function mutations.

The existence of completely dispensable interactions is confirmed by in vitro experiments based on yeast two-hybrid assays[8]. In addition, genome-wide analysis suggests widespread occurrence of completely dispensable interactions in protein phosphorylation[26,27]. Using the PANTHER[57] webtool for Gene Ontology analysis of dispensable interactions identified by both experiments and predictions, we found that none of the Gene Ontology terms are significantly enriched in completely dispensable interactions (false discovery rate <0.05), consistent with the expectation that completely dispensable interactions tend to be non-functional or not well-studied in the literature.

Fitness measurements under laboratory conditions do not accurately reflect selective pressures in natural environments over evolutionary timescales[58–60]. Hence, instead of using fitness measurements under laboratory conditions, we use population genetic datasets to accurately measure selective pressures and fitness effects of mutations. Another important factor to consider is macromolecular crowding, which is known to modulate protein–protein interactions in vivo[61]. In this study, we make the reasonable assumption that macromolecular crowding exerts similar thermodynamic effects on each binary protein–protein interaction before and after mutation. Crowding effects can be modulated by several factors, including protein shape[61]. The effects of protein shape on crowding at the interactome scale remains to be investigated in future work.

In summary, we estimate that up to ~20% of the overall human interactome is completely dispensable. This estimate represents an average over the entire human interactome, likely with significant variations within the interactome. Indeed, dispensable PPI content may be much larger in certain subsets of the interactome, specifically transient PPIs mediated by motif-domain interactions[25,27]. Our study suggests that the majority of the human interactome is under strong purifying selection, enabling the maintenance of a somewhat close-to-streamlined interactome (where non-dispensable interactions outnumber completely dispensable interactions) in the presence of mutation and genetic drift. Furthermore, our study provides a solid justification for the utility of interactome networks in elucidating the phenotypic consequences of genetic mutations. These insights are enabled by systematic determination of precise interactome perturbation patterns induced by mutations, and they illustrate the power and utility of complementing high-resolution mutation edgotyping experiments with structural systems biology computations.

## Methods

**Construction of the human structural interactome**. Three-dimensional (3D) protein structures at atomic resolution were retrieved in October 2017 from the Protein Data Bank (PDB)[45]. For structures containing more than one model, the first model was selected. Gene names and gene Entrez IDs in the HI-II-14 reference interactome were mapped to protein UniProt IDs and corresponding amino acid sequences using the ID mapping table provided by UniProt[62]. For proteins in the IntAct reference interactome, UniProt IDs provided by the IntAct database were used to obtain corresponding amino acid sequences. Next, we used BLAST[63] to perform sequence alignment on all protein sequences against all PDB chain sequences found in PDB's SEQRES records, with an E-value cut-off of $10^{-10}$. For each pair of protein sequence and PDB chain, the alignment with the smallest E-value was retained, and the remaining alignments were discarded. A PPI was annotated with a pair of interacting chains in the same PDB structure (with at least one interface residue mediating the interaction) if (i) one of the proteins in the PPI has a sequence alignment with one of the chains in the chain pair, with ≥50% of interface residues mapped onto the protein; and (ii) the other protein in the PPI has a sequence alignment with the other chain in the chain pair, with ≥50% of interface residues mapped onto the protein. PPIs without any PDB chain-pair annotations were discarded. For each structurally annotated PPI, up to five PDB chain-pair annotations with the smallest joint alignment E-values were used to identify interface residues, and the rest chain-pair annotations were discarded.

**Identifying binding interface residues for two chains in a PDB structure**. 3D coordinates at atomic resolution for each chain were loaded from the PDB structure using the Biopython library[64], and amino acid residues associated with these coordinates were verified with the chain's backbone sequence provided by the SEQRES records of PDB. Residues that are not part of the chain's backbone sequence were discarded. Next, we calculated the Euclidean distance between each residue of one chain and all residues of the other chain. The distance between two residues was calculated as the minimum distance between all atoms of the first residue and all atoms of the second residue. If the residue of one chain is within a distance of 5 Å from any residue in the other chain, that residue was labeled as an interface residue.

**Mapping disease mutations onto the human structural interactome**. Germline mutations in human with associated phenotypic consequences were retrieved in February 2019 from the ClinVar database[46] (genome assembly GRCh38). We selected missense mutations that are strictly labeled as pathogenic only, with supporting evidence (i.e., with at least one star), and with no conflicting phenotypic interpretations. To map mutations onto proteins in the human structural interactome, we searched the protein's RefSeq transcript provided by ClinVar for the mutation flanking sequence, defined as either the first 10 amino acid residues or all amino acid residues, whichever one is shorter, on both sides of the mutation. Then we searched the protein's sequence designated by UniProt for the mutation flanking sequence obtained from the RefSeq transcript. If the flanking sequence was found on the protein sequence at the same position reported by ClinVar, the mutation was retained for further analysis, otherwise the mutation was discarded. For multiple mutations mapping onto the same position, only one mutation was retained for further analysis.

**Mapping non-disease mutations onto the human structural interactome**.
Single-nucleotide polymorphism (SNP) mutations in human were retrieved in October 2017 from the Single Nucleotide Polymorphism Database (dbSNP)[47] (build 150 GRCh38p7). First, we selected only missense SNPs that are labeled as validated and not withdrawn, and are assigned a location on the RefSeq transcript of a protein. Next, we discarded all mutations labeled with disease assertions (e.g., pathogenic, likely pathogenic, drug-response, uncertain significance or other). Then we selected mutations whose minor allele frequencies are higher than 1%, as common mutations with high frequencies are unlikely to be associated with a disease. To map mutations onto proteins in the human structural interactome, we searched the protein's RefSeq transcript provided by dbSNP for the mutation flanking sequence, defined as either the first 10 amino acid residues or all amino acid residues, whichever one is shorter, on both sides of the mutation. Then we searched the protein's sequence designated by UniProt for the mutation flanking sequence obtained from the RefSeq transcript. If the flanking sequence was found on the protein sequence at the same position reported by dbSNP, the mutation was retained for further analysis, otherwise the mutation was discarded. Finally, mutations overlapping in position with disease mutations were also discarded.

**Calculating functional similarity between two proteins**. Gene Ontology (GO) associations were retrieved in March 2019 from the Gene Ontology Consortium[65,66], which provides a set of controlled hierarchical GO terms distributed among three root categories: ~29,600 biological process terms, ~11,100 molecular function terms, and ~4200 cellular component terms. Functional similarity between two proteins was then calculated using the SimGIC[67] semantic similarity measure implemented in the Fastsemsim python library.

**Calculating tissue co-expression for two proteins**. Gene tissue expression data were retrieved from four databases: the Illumina Body Map 2.0 project[68] with RNA-seq data in 16 normal human body tissues (log2 transformed), the Genotype-Tissue Expression (GTEx) project[69] with normalized RNA-seq data in 48 normal human body tissues, the Human Protein Atlas (HPA)[70] with protein immunohistochemistry microarray data in 44 normal human body tissues, and the Fantom5 project[71] with CAGE (Cap Analysis of Gene Expression) peaks (tags per million) for gene promoters in 183 normal human body tissue samples. For GTEx data, gene expression levels in each tissue were averaged over all samples. For HPA data, gene expression levels were mapped from the four symbolic values {not detected, low, medium, high} to numeric values {0, 1, 2, 3}, respectively. For Fantom5 data, promoter CAGE peaks were mapped to genes using the associated HGNC IDs. For genes with multiple CAGE peaks, the average over all peaks was considered. Tissue co-expression for two proteins was then calculated using Pearson's correlation coefficient for their tissue expression profiles. Only protein pairs whose expression levels are defined together in at least five tissues were considered.

**Calculating the 95% confidence interval of the fraction of completely dispensable PPIs**. Each mutation can be either edgetic (E) or not edgetic (E). In addition, the fitness effect of a mutation can be either neutral (N), mildly deleterious (M), or severely detrimental (S). The fraction of PPIs effectively neutral upon edgetic disruption P(N|E) was calculated using Bayes' theorem: $P(N|E) = P(E|N)P(N)/P(E)$, where $P(E) = P(E|N)P(N) + P(E|M)P(M) + P(E|S)P(S) = P(E|N)P(N) + P(E|M)P(M)$, assuming that $P(E|S) = 0$. Since the probabilities P(N) and P(M) are constants, it is easy to see that P(N|E) only depends on P(E|M)/P(E|N) in the following way: $1/P(N|E) = \{P(E|M)/P(E|N)\} \times \{P(M)/P(N)\} + 1$. The 95% confidence interval for the ratio of two proportions P(E|M)/P(E|N) was calculated according to Bland[72], which was then used to calculate the 95% confidence interval for P(N|E) using the above equation.

## Data availability

The human structural interactomes (Y2H-SI and IntAct-SI) and genetic mutations analyzed in this study are included in this article and its Source Data files. The HI-II-14 reference interactome is available at The Human Reference Protein Interactome Mapping Project (http://interactome.baderlab.org). The IntAct reference interactome is available at the IntAct Molecular Interaction Database (http://www.ebi.ac.uk/intact). Protein sequences are available at the UniProt database (https://www.uniprot.org). Three-dimensional structural templates used for the modeling of protein–protein interactions are available at the Protein Data Bank (https://www.wwpdb.org). Non-disease missense mutations are available at the dbSNP database (https://www.ncbi.nlm.nih.gov/snp). Disease-causing missense mutations are available at the ClinVar database (https://www.ncbi.nlm.nih.gov/clinvar). Gene ontology association data underlying supplementary figures are available at the Gene Ontology database (http://geneontology.org). Gene tissue expression data underlying supplementary figures are available at the Illumina Body Map 2.0 project (https://www.ebi.ac.uk/gxa/experiments/E-MTAB-513), the Genotype-Tissue Expression (GTEx) project (https://gtexportal.org/home/datasets), the Human Protein Atlas (HPA) project (https://www.proteinatlas.org/about/download) and the Functional Annotation Of The Mammalian Genome (FANTOM5) project (http://fantom.gsc.riken.jp/5/datafiles/reprocessed/hg38_latest/extra). The source data underlying Figs. 2, 3, and 4b–e are provided as Supplementary Data files.

## Code availability

Software code used for data analyses and calculations is available at https://github.com/MohamedGhadie/dispensable_ppi_content.

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

## Acknowledgements

This work was supported by Natural Sciences and Engineering Research Council of Canada grants RGPIN-2019-05952 and RGPAS-2019-00012, Canada Foundation for Innovation grants JELF-33732 and IF-33122, and Canada Research Chairs program to Y.X., and McGill Engineering Doctoral Awards program to M.G. The funders had no role in study design, data collection and analysis, decision to publish, or preparation of the paper.

## Author contributions

Y.X. conceived and oversaw all aspects of the project. M.G. and Y.X. designed experiments. M.G. performed experiments and analyzed data. Y.X. supervised research. M.G. and Y.X. wrote the paper.

## Additional information

**Competing interests:** The authors declare no competing interests.

