## [Peer Review File · Nature Communications]

Reviewers' comments:

Reviewer #1 (Remarks to the Author):

Xia et al. performed junk content estimation in the human interactome by calculating the fractions of PPIs perturbed by mutations. Although this is an interesting topic and the analyses are well designed, several issues need to be addressed before publication.

Major:

- 1) The authors evaluated the functional and tissue co-expression of the structural interactome based on GO terms and gene expression. First, the Jaccard index is not a good way to measure the functional similarity, the relationship between GO terms needs to be considered. Second, only Human Body Map expression data were used, other datasets across multiple tissues should be used to validate the conclusions, such as GTEx, HPA and Fantom5 projects. Third, the authors compared their findings to only the reference interactome, how about the comparison with random interactions between protein pairs?
- 2) The authors estimate the junk based on Y2H-SI and IntAct-SI separately, do the findings change if they combine these two networks?
- 3) The authors declared that the PPI is considered perturbed by the mutation if and only if $\Delta G > 0.5$ kcal/mol. It is not clear which mutations are considered in the analysis, only the mutations that decrease the ΔG or the ones that both decrease/increase are considered. Some mutations might lose PPIs and some mutations can also enhance PPIs, the perturbation is defined as loss of interactions?
- 4) The disease mutations were obtained from HGMD 2011.4, this is a very old dataset. The updated data need to be used to validate their conclusions.

Minor:

- 1) Some figures can be combined together, such as Figure-1 and Figure-2; Figure-5 and Figure-6;
- 2) Only BindProfX was used to predict the ΔG , other methods need to be included, such as STRUM or FoldX.

Reviewer #2 (Remarks to the Author):

In "Estimating junk content in the human interactome", Ghadie and Xia pose an intriguing question, namely that of protein interactions which appear to be "inert" to mutations. Unfortunately it appears that several important aspects are not considered in the current version of the manuscript.

"Junk" interactions does not appear to be an established term; I have not come across it, and Google only shows hits to a previous paper by the same authors. Perhaps they mean what's commonly referred to as "non-specific interactions" (see e.g. <https://doi.org/10.1038/nature05306>, and may well be functionally relevant for the cell). Another possibility would be to consider these robust proteins/interactions (in the sense of being tolerant to mutations) - evolution is an expert at "engineering"/optimising for tolerance, after all.

A complicating factor are gain-of-function mutations - a prominent example is sickle cell anaemia, where the mutation leads to a "bump" on the protein which then leads to stacking of hemoglobins, in a way these proteins normally would not interact. Such gain-of-function variants are generally considered to be among the hardest to predict, and - as the gained binding partner isn't known a priori - are also extremely hard to screen for. So far they are mostly identified serendipitously. This is not an aspect the authors' model. Given all the constraints outlined above, it may be hard to systematically account for gain-of-function variants, but at the very least the authors should acknowledge this challenge, and discuss possible implications.

Overall, I would suggest that the authors incorporate literature on non-specific interactions, robustness of proteins, and gain-of-function mutations, and outline how their proposed subset of interactions fit into this framework.

On the technical side, mapping of interactomes and disease variants to structure has been done

before (e.g. <https://doi.org/10.1038/nmeth.2289>,
<https://dx.doi.org/10.1093%2Fbioinformatics%2Fbtt181>, <https://doi.org/10.1038/nmeth.3289>). A somewhat surprising aspect is that the authors have near-identical numbers of common and disease-associated variants, though generally truly common & benign variants (>1% population frequency) are rather rare, while disease-associated missense variants are common. Standard in the field is now to use ClinVar with "at least one star" for high-quality disease-associated variants.

RESPONSE TO COMMENTS FROM REVIEWER #1

Reviewer's Comments: Xia *et al.* performed junk content estimation in the human interactome by calculating the fractions of PPIs perturbed by mutations. Although this is an interesting topic and the analyses are well designed, several issues need to be addressed before publication.

Author's Response: We thank the reviewer for the positive evaluation of our work and for the very insightful comments, which helped us refine and improve our manuscript. We have addressed all the reviewer's comments as described below.

Reviewer's Comments: (1) The authors evaluated the functional and tissue co-expression of the structural interactome based on GO terms and gene expression. First, the Jaccard index is not a good way to measure the functional similarity, the relationship between GO terms needs to be considered.

Author's Response: We thank the reviewer for this important suggestion to accurately measure GO-based functional similarity among protein-protein interaction partners by considering the relationship between GO terms. Thus, we have repeated our calculations with the widely used GO-based semantic similarity measure SimGIC (Pesquita *et al.*, Metrics for GO based protein semantic similarity: a systematic evaluation. *BMC Bioinformatics* 9(Suppl 5):S4, 2008). This measure considers the information content of GO terms when calculating GO-based functional similarity, and is considered one of the best measures for assessing protein-protein interactions (Pesquita *et al.*, *PLoS Comput. Biol.* 5:e1000443, 2009; Guzzi *et al.*, *Brief. Bioinform.* 13:569-585, 2012). Using this measure for GO-based functional similarity, our conclusions remain the same: interaction partners in our structural interactomes exhibit significantly higher functional similarity compared to interaction partners in the reference interactome as well as compared to random protein pairs, thus confirming the high quality of our structural interactomes. We have revised our manuscript accordingly to include these new results (Page 5 Lines 91-95; Page 21 Lines 417-422; Page 29 Lines 683-687; Figure S1).

Reviewer's Comments: (2) Second, only Human Body Map expression data were used, other datasets across multiple tissues should be used to validate the conclusions, such as GTEx, HPA and Fantom5 projects.

Author's Response: We thank the reviewer for this important suggestion. We have additionally validated our conclusions using data from GTEx, HPA and Fantom5 projects as suggested by the reviewer. Using these three new datasets, our conclusions remain the same: interaction partners in our structural interactomes exhibit significantly higher tissue co-expression compared to interaction partners in the reference interactome as well as compared to random protein pairs, thus confirming the high quality of our structural interactomes. We have revised our manuscript accordingly to include these new results (Page 5 Lines 91-95; Pages 21-22 Lines 423-437; Page 29 Lines 678-682; Figure S1).

Reviewer's Comments: (3) Third, the authors compared their findings to only the reference interactome, how about the comparison with random interactions between protein pairs?

Author's Response: We thank the reviewer for this important suggestion. In the revised manuscript, we have compared GO-based functional similarity and tissue co-expression of interaction pairs in our structural interactomes to those in the reference interactome as well as to random protein pairs. Our comparisons show that interaction partners in our structural interactomes exhibit significantly higher functional similarity and tissue co-expression compared to interaction partners in the reference interactome as well as compared to random protein pairs, thus confirming the high quality of our structural interactomes. We have revised our manuscript accordingly to include these new results (Page 5 Lines 91-95; Page 29 Lines 677-687; Figure S1).

Reviewer's Comments: (4) The authors estimate the junk based on Y2H-SI and IntAct-SI separately, do the findings change if they combine these two networks?

Author's Response: We thank the reviewer for this very insightful question. We did not perform this calculation in the initial submission because 87% of Y2H-SI is already part of the literature-curated IntAct-SI. In the revised manuscript, we have repeated our junk PPI calculations using the combined network of Y2H-SI and IntAct-SI. Our geometry-based calculations showed that 18.1% of the combined interactome is junk, with a 95% confidence interval of 15.2% to 21.4%. In addition, our physics-based calculations showed that 15.8% of the combined interactome is junk, with a 95% confidence interval of 12.6% to 19.6%. These results are in broad agreement with our main conclusion that $\sim 20\%$ of the human interactome is junk. We have revised our manuscript accordingly to include these new results (Page 13 Lines 231-233).

Reviewer's Comments: (5) The authors declared that the PPI is considered perturbed by the mutation if and only if $\Delta\Delta G > 0.5$ kcal/mol. It is not clear which mutations are considered in the analysis, only the mutations that decrease the $\Delta\Delta G$ or the ones that both decrease/increase are considered. Some mutations might lose PPIs and some mutations can also enhance PPIs, the perturbation is defined as loss of interactions?

Author's Response: We thank the reviewer for bringing up this important conceptual point and for giving us the opportunity to clarify our physics-based approach for predicting mutation-induced PPI perturbations. In this study, we are only interested in the fraction of all mutations that eliminate pre-existing PPIs in the reference interactome (i.e., $\Delta\Delta G > 0.5$ kcal/mol). This is because our goal is to estimate the fraction of PPIs in the reference interactome that are effectively neutral upon disruption or elimination only. It is possible for mutations to enhance pre-existing PPIs by increasing binding affinity (i.e., $\Delta\Delta G < 0$ kcal/mol), but they are not considered to be PPI-disrupting, as they maintain rather than eliminate existing PPIs. We have revised our manuscript accordingly to better clarify our physics-based approach (Pages 11-12, Lines 185-200; Page 12, Lines 204-207). In addition, we have revised the definition of "junk" PPI from "neutral upon perturbation" to the more precise "neutral upon disruption" throughout the manuscript.

Reviewer's Comments: (6) The disease mutations were obtained from HGMD 2011.4, this is a very old dataset. The updated data need to be used to validate their conclusions.

Author's Response: We thank the reviewer for this excellent suggestion which helped us refine and improve our manuscript. We have updated our dataset of disease mutations to the most

recent data from the ClinVar database (as requested by Reviewer #2), which was retrieved in February 2019. We selected only high-quality missense mutations strictly labeled as “pathogenic” only, with supporting evidence (i.e., with at least one star), and with no conflicting interpretations. We repeated our analysis using this high-quality dataset, and all of our conclusions remain unchanged. In particular, we still conclude that up to ~20% of the human interactome is junk, suggesting that the majority of the interactome is under strong purifying selection. We have modified the manuscript accordingly to include all updated results based on this most recent dataset.

Reviewer’s Comments: (7) Some figures can be combined together, such as Figure-1 and Figure-2; Figure-5 and Figure-6.

Author’s Response: We thank the reviewer for the important recommendation which helped us improve our manuscript. We have combined Figure 1 and Figure 2 into one figure (Figure 1), and also combined Figure 5 and Figure 6 into one figure (Figure 4).

Reviewer’s Comments: (8) Only BindProfX was used to predict the $\Delta\Delta G$, other methods need to be included , such as STRUM or FoldX.

Author’s Response: We thank the reviewer for this important suggestion which helped us improve our manuscript. In the initial submission, we chose BindProfX over STRUM because while both methods were developed by the Zhang Lab at the University of Michigan, BindProfX is the most recent method specifically designed for handling protein-protein interactions. Following the reviewer’s suggestion, in the revised manuscript, we have repeated our physics-based predictions of mutation-induced PPI perturbations using $\Delta\Delta G$ values calculated by FoldX, and re-calculated the fraction of junk PPIs in both Y2H-SI and IntAct-SI using these new FoldX-based PPI perturbation predictions. In Y2H-SI, we found that 11.1% of the human interactome is junk, with a 95% confidence interval of 4.7% to 24.1%. In IntAct-SI, we found that 13% of the human interactome is junk, with a 95% confidence interval of 9.5% to 17.5%. These estimates remain in broad agreement with our estimates obtained from $\Delta\Delta G$ values calculated by BindProfX (12.5% in Y2H-SI, and 16% in IntAct-SI), and further confirm our main conclusion that <~20% of the human interactome is junk. The manuscript has been revised accordingly to include these new results (Page 15 Lines 269-275; Page 30 Lines 728-735).

RESPONSE TO COMMENTS FROM REVIEWER #2

Reviewer’s Comments: In “Estimating junk content in the human interactome”, Ghadie and Xia pose an intriguing question, namely that of protein interactions which appear to be “inert” to mutations. Unfortunately it appears that several important aspects are not considered in the current version of the manuscript.

Author’s Response: We thank the reviewer for the positive evaluation of our work and for the very insightful comments, which helped us refine and improve our manuscript.

Reviewer’s Comments: (1) “Junk” interactions does not appear to be an established term; I have not come across it, and Google only shows hits to a previous paper by the same authors. Perhaps

they mean what's commonly referred to as “non-specific interactions” (see e.g. <https://doi.org/10.1038/nature05306>, and may well be functionally relevant for the cell). Another possibility would be to consider these robust proteins/interactions (in the sense of being tolerant to mutations) - evolution is an expert at “engineering”/optimising for tolerance, after all.

Author's Response: We thank the reviewer for bringing up this very important point and for giving us the opportunity to clarify the aim of our work. The term “junk” has long been used in the context of DNA (Ohno, *Brookhaven Symp. Biol.* 23:366-370, 1972) with a precise and well-defined meaning (i.e., “junk” DNA is defined as DNA sites that are neutral upon disruption). We agree with the reviewer that we may be the first to use this term in the context of protein-protein interaction while preserving its original meaning (i.e., “junk” interactions are similarly defined as protein-protein interactions that are neutral upon disruption). The reason we choose to use the term “junk” is that no other existing term can accurately capture the meaning of “neutral upon disruption”, including the term “non-specific” interaction which has been used interchangeably in the literature to refer to different types of interactions with no agreed-upon definition. For example, the term “non-specific” interactions has been used in the literature to describe non-stereospecific interactions or transient interactions that may well be crucial to cellular function (Blundell and Fernandez-Recio, *Nature* 444:279-280, 2006; Tang *et al.*, *Nature* 444:383-386, 2006), interactions that involve promiscuous binding of a protein to many partners (Schreiber and Keating, *Curr. Opin. Struct. Biol.* 21:50-61, 2011), or interactions that may have no function (Kanshin *et al.*, *Cell Rep.* 10:1202-1214, 2015). While the last definition of “non-specific” interactions comes close to our definition of “junk” interactions, the first two definitions of “non-specific” interactions are very different from our definition of “junk” interactions. It is possible for “non-specific” interactions to be not “junk” if they are important to cellular function and deleterious upon disruption. Conversely, it is possible for “specific” interactions to be “junk” if their disruption does not affect organismal fitness. While “junk” interactions may or may not be associated with well-defined molecular functions, what uniquely defines them is that they do not measurably affect organismal fitness upon disruption. Therefore, it is important for us to make a clear distinction between “junk” interactions (which are neutral upon disruption) and “non-specific” interactions (which may or may not be neutral upon disruption).

We completely agree with the reviewer that “junk” interactions can lead to increasing robustness of cellular networks against mutations. However, we are hesitant to use the term “robust” interactions, because it can refer to two very different types of interactions. On the one hand, “robust” interactions can refer to stable interactions that are preserved in the presence of many mutations at the binding interface (Vo *et al.*, *Cell* 164:310-323, 2016; Leducq *et al.*, *PLoS Genet.* 8:e1003161, 2012; Ghadie *et al.*, *Curr. Opin. Struct. Biol.* 50:42-48, 2018). On the other hand, “robust” interactions can also refer to interactions whose elimination by mutations does not induce any phenotypic change (Jubb *et al.*, *Prog. Biophys. Mol. Biol.* 128:3-13, 2017; Sahni *et al.*, *Cell* 161:647-660, 2015; Yates and Sternberg, *J. Mol. Biol.* 425:3949-3963, 2013; Ghadie *et al.*, *Curr. Opin. Struct. Biol.* 50:42-48, 2018). While the second definition of “robust” interactions comes close to our definition of “junk” interactions, the first definition of “robust” interactions is very different from our definition of “junk” interactions. Indeed, the first definition of “robust” interactions may well be not “junk”, if these stable interactions (which are preserved under mutation) are important to cellular function and deleterious upon elimination. Our usage of the term “junk” interactions minimizes such confusion, as it is much more specific than the term “robust” interactions.

We have revised our manuscript accordingly to highlight the ability of “junk” interactions to increase robustness of cellular networks against mutations (Page 3 Lines 50-55). In addition, we have included the relevant literature citations on “non-specific” interactions and “robust” interactions in order to clearly distinguish them from our definition of “junk” interactions (Page 2 Lines 31-40; Page 3 Lines 50-55).

Reviewer’s Comments: (2) A complicating factor are gain-of-function mutations - a prominent example is sickle cell anaemia, where the mutation leads to a “bump” on the protein which then leads to stacking of hemoglobins, in a way these proteins normally would not interact. Such gain-of-function variants are generally considered to be among the hardest to predict, and - as the gained binding partner isn’t known a priori - are also extremely hard to screen for. So far they are mostly identified serendipitously. This is not an aspect the authors’ model. Given all the constraints outlined above, it may be hard to systematically account for gain-of-function variants, but at the very least the authors should acknowledge this challenge, and discuss possible implications. Overall, I would suggest that the authors incorporate literature on non-specific interactions, robustness of proteins, and gain-of-function mutations, and outline how their proposed subset of interactions fit into this framework.

Author’s Response: We thank the reviewer for bringing up this very important point which helped us refine and improve our manuscript, and for giving us the opportunity to clarify the scope of our work. We completely agree with the reviewer that gain-of-function mutations represent an exciting frontier of research that needs to be explored in the context of molecular interactions and interactome networks (Yates and Sternberg, *J. Mol. Biol.* 425:3949-3963, 2013; Li and Babu, *Cell* 175:40-42, 2018). In addition to the classic sickle cell anaemia example given by the reviewer, many specific examples of gain-of-interaction mutations have been identified that are important in cancer (van Oijen and Slootweg, *Clin. Cancer Res.* 6:2138-2145, 2000; Kakiuchi *et al.*, *Nat. Genet.* 46:583-587, 2014), neurodegenerative diseases (Lashuel *et al.*, *Biochemistry* 38:13560-13573, 1999), as well as other diseases (Meyer *et al.*, *Cell* 175:239-253, 2018). Despite many such case studies on gain-of-interaction mutations, we do not yet know for certain how often they occur at the genomic scale. According to the recent genome-wide experimental study of mutation edgotypes (Sahni *et al.*, *Cell* 161:647-660, 2015), only ~1% (2 out of 197) of disease mutations confer gain of interaction, compared to ~30% of disease mutations that confer edgetic loss of interaction. Thus, gain-of-interaction mutations appear to be ~30 times less likely to occur than edgetic loss-of-interaction mutations in human disease. More work is clearly needed in the future to confirm these preliminary results in the literature.

The presence of gain-of-function mutations does not affect our estimate of “junk” content in the reference interactome. Our definition of “junk” interactions only refers to pre-existing PPIs in the reference interactome which are neutral upon elimination by mutation. Hence, our definition of “junk” interactions is independent of the extent of gain-of-function mutations. Furthermore, our Bayesian formulation for estimating junk PPI content, $P(N|E) = P(E|N)P(N) / \{P(E|N)P(N) + P(E|M)P(M) + P(E|S)P(S)\}$, is robust to gain-of-function mutations. The three prior probabilities $P(N)$, $P(M)$ and $P(S)$, for new missense mutations to be neutral (N), mildly deleterious (M) and strongly detrimental (S), are obtained from the literature using procedures that are robust to gain-of-function mutations (Kryukov *et al.*, *Am. J. Hum. Genet.* 80:727-739, 2007). In addition, the three conditional probabilities $P(E|N)$, $P(E|M)$ and $P(E|S)$, for neutral (N), mildly deleterious (M) and strongly detrimental (S) mutations to edgetically eliminate PPIs (E), are also independent of the extent of gain-of-function mutations. Therefore, our “junk” PPI estimate $P(N|E)$ does not depend on the extent of gain-of-function mutations.

Since our goal in this study is to estimate “junk” content among pre-existing PPIs in the reference interactome only, gain-of-function mutations are beyond the scope of our current study. At the same time, our Bayesian framework can be extended in the future to the calculation of “junk” content in *de novo* PPIs newly created by gain-of-function mutations.

We have revised our manuscript accordingly and included the relevant literature citations on gain-of-function mutations (Pages 17-18 Lines 323-344), non-specific interactions (Page 2 Lines 31-40), and robust interactions (Page 3 Lines 50-55), as well as how they relate to our estimate of junk content in the reference interactome.

Reviewer’s Comments: (3) On the technical side, mapping of interactomes and disease variants to structure has been done before (e.g. <https://doi.org/10.1038/nmeth.2289>, <https://dx.doi.org/10.1093%2Fbioinformatics%2Fbtt181>, <https://doi.org/10.1038/nmeth.3289>). A somewhat surprising aspect is that the authors have near-identical numbers of common and disease-associated variants, though generally truly common & benign variants (>1% population frequency) are rather rare, while disease-associated missense variants are common. Standard in the field is now to use ClinVar with “at least one star” for high-quality disease-associated variants.

Author’s Response: We thank the reviewer for these very important suggestions. In the revised manuscript, we have included the references suggested by the reviewer as well as other references on prior work of mapping interactomes and disease variants to 3D structure (Page 3 Lines 64-66). In addition, following the reviewer’s suggestion, we have updated our dataset of disease mutations to the most recent data from the ClinVar database, which was retrieved in February 2019. We selected only high-quality missense mutations strictly labeled as “pathogenic” only, with supporting evidence (i.e., with at least one star), and with no conflicting interpretations. Since the quality of disease-causing mutation annotations is of utmost importance to us, we decided to exclude other mutations that may not be strictly disease-causing, including but not limited to those with labels such as “likely pathogenic”, “association”, “drug response” and “affects”. We repeated our analysis with this high-quality dataset of disease-causing mutations, and all of our conclusions remain unchanged. In particular, we still conclude that up to ~20% of the human interactome is junk, suggesting that the majority of the interactome is under strong purifying selection.

The small size of our disease-causing mutation dataset (relative to our common mutation dataset) simply reflects the far-from-complete nature of the current understanding of human disease, as well as our stringent criteria for selecting disease-causing mutations as described above. It is important to note that our Bayesian framework for junk PPI calculation never makes use of the sizes of our disease-causing and common mutation datasets. Rather, we only use our disease-causing and common mutation datasets to calculate the conditional probabilities $P(E|N)$ and $P(E|M)$ for neutral (N) and mildly deleterious (M) mutations to be edgetic (E), which are completely independent of the sizes of our disease-causing and common mutation datasets. In addition, our Bayesian framework for junk PPI calculation also makes use of the literature-derived prior probability for new missense mutations to be neutral (N): $P(N) = 0.27$ (Kryukov *et al.*, *Am. J. Hum. Genet.* 80:727-739, 2007). This literature-derived prior probability suggests that ~73% of new missense mutations are deleterious and only ~27% of new missense mutations are effectively neutral, in agreement with the reviewer’s expectation that disease-causing mutations are more common than neutral mutations.

In response to the reviewer’s suggestions, the manuscript has been modified accordingly to include all updated results based on the most recent data from the ClinVar database.

REVIEWERS' COMMENTS:

Reviewer #1 (Remarks to the Author):

the authors have answered and addressed in detail all my questions.

Reviewer #2 (Remarks to the Author):

Ghadie and Xia have submitted a revised manuscript that addresses many of the reviewers' comments. It also raises some new issues, specifically regarding the use of FoldX. While it is a useful tool in several contexts, it has not been benchmarked for the use of homology models (or if it has been, the authors should cite performance and the underlying paper). The authors would need to benchmark its performance based on e.g. the SKEMPI dataset, or another set geared towards protein interactions. Alternatively, the authors can restrict the FoldX analysis to interactions for which co-crystal structures are available.

The term 'junk' should not be used. It has been abandoned in the field of genetics for many years. Anecdotally, the title of Mike Eisen's blog is "it is NOT junk".

It does not become clear to this referee how robust interactions are different from the subset they discuss. Please elaborate.

Lastly, I urge the authors to discuss the biological meaning of these interactions. Do you have any example where this has been confirmed in vivo, or even in vitro? The Bolon lab and later Fraser lab ubiquitin deep mutational scanning papers show that almost all ubiquitin residues can be mutated and the protein is still functional - nevertheless evolution has strongly conserved it. Thus, while interesting in many cases, conclusions from individual mutagenesis experiments may not allow direct interpretation - especially if those conclusions show a lack of an effect. The dynamic range in lab assays is very different from what evolution picks up on. To place this paper in the context of ongoing research, it is critical to let readers know what the functional implications may be. It would also be extremely interesting if there were cases of such interactions that have been described in the literature based on experimental observation. In this context, the authors may want to consider this recently published article: <https://www.pnas.org/content/115/43/10965>

RESPONSE TO COMMENTS FROM REVIEWER #1

Reviewer's Comments: the authors have answered and addressed in detail all my questions.

Author's Response: We thank the reviewer for the positive evaluation of our work.

RESPONSE TO COMMENTS FROM REVIEWER #2

Reviewer's Comments: Ghadie and Xia have submitted a revised manuscript that addresses many of the reviewers' comments.

Author's Response: We thank the reviewer for the positive evaluation of our work.

Reviewer's Comments: It also raises some new issues, specifically regarding the use of FoldX. While it is a useful tool in several contexts, it has not been benchmarked for the use of homology models (or if it has been, the authors should cite performance and the underlying paper). The authors would need to benchmark its performance based on e.g. the SKEMPI dataset, or another set geared towards protein interactions. Alternatively, the authors can restrict the FoldX analysis to interactions for which co-crystal structures are available.

Author's Response: We thank the reviewer for this important suggestion. We have benchmarked the performance of FoldX for the use of homology models, using the SKEMPI dataset as the gold standard. We found that the performance of FoldX using homology models is only slightly worse than using co-crystal structures (Pearson correlation coefficient between predicted versus experimental $\Delta\Delta G$ is 0.50 for co-crystal structures, and 0.42 for homology models), with the crucial benefit of significantly increasing interactome coverage (17 co-crystal structures versus 469 homology models in the HI-II-14 interactome, and 158 co-crystal structures versus 3175 homology models in the IntAct interactome), which is essential for our interactome-scale calculations. Finally, by integrating co-crystal structures with homology models, FoldX-based calculations completely agree with three other methods (geometry-based calculations, BindProfX-based calculations, and yeast two-hybrid experiments) in supporting our principal conclusion that <20% of the human interactome is neutral upon disruption. We have revised our manuscript accordingly to include these new calculations (Page 16 Lines 281-286).

Reviewer's Comments: The term 'junk' should not be used. It has been abandoned in the field of genetics for many years. Anecdotally, the title of Mike Eisen's blog is "it is NOT junk".

Author's Response: We thank the reviewer for this important suggestion. As suggested by the reviewer and by the editor, we have replaced the term "junk" with the term "dispensable" throughout the manuscript.

Reviewer's Comments: It does not become clear to this referee how robust interactions are different from the subset they discuss. Please elaborate.

Author's Response: We thank the reviewer for giving us the opportunity to clarify why we prefer the term "dispensable interactions" over the term "robust interactions". The term "dispensable

interactions” has a clear and specific definition. It refers to molecular interactions that are neutral to the organism upon disruption. In other words, the fitness level of the organism remains the same as the wild type when these interactions are knocked out from the interactome. In contrast, the term “robust interactions” is highly ambiguous and can refer to two completely different types of molecular interactions. On the one hand, “robust interactions” can refer to interactions whose elimination by mutations does not induce any phenotypic change, similar to our definition of dispensable interactions. On the other hand, “robust interactions” can also refer to stable interactions that are preserved in the presence of mutations at the binding interface, which is completely different from our definition of dispensable interactions. Indeed, interactions that are stable in the presence of mutations are frequently not dispensable, if they are important to cellular function and hence deleterious upon elimination. Our usage of the term “dispensable interactions” minimizes such confusion, as it is clearer and more specific than the term “robust interactions”. We have revised our manuscript accordingly to clarify these points (Page 3 Lines 52-58).

Reviewer’s Comments: Lastly, I urge the authors to discuss the biological meaning of these interactions. Do you have any example where this has been confirmed *in vivo*, or even *in vitro*? The Bolon lab and later Fraser lab ubiquitin deep mutational scanning papers show that almost all ubiquitin residues can be mutated and the protein is still functional - nevertheless evolution has strongly conserved it. Thus, while interesting in many cases, conclusions from individual mutagenesis experiments may not allow direct interpretation - especially if those conclusions show a lack of an effect. The dynamic range in lab assays is very different from what evolution picks up on. To place this paper in the context of ongoing research, it is critical to let readers know what the functional implications may be. It would also be extremely interesting if there were cases of such interactions that have been described in the literature based on experimental observation. In this context, the authors may want to consider this recently published article: <https://www.pnas.org/content/115/43/10965>

Author’s Response: We thank the reviewer for the excellent suggestion of discussing the biological meaning of dispensable interactions. Indeed, *in vitro* experiments based on yeast two-hybrid assays have confirmed 26 dispensable interactions which are eliminated by common mutations in healthy individuals (Sahni *et al.*, *Cell* 161:647-60, 2015). Using the PANTHER webtool for Gene Ontology analysis of dispensable interactions identified by both experiments and calculations (Mi *et al.*, *Nucleic Acids Res.* 41:D377-86, 2013), we found that none of the Gene Ontology terms are significantly enriched in dispensable interactions (false discovery rate < 0.05), consistent with the expectation that dispensable interactions tend to be non-functional or not well-studied in the literature. The only other focused study of dispensable interactions in the literature is done by the Michnick lab, who has found widespread occurrence of dispensable interactions in protein phosphorylation (Levy *et al.*, *Phil. Trans. R. Soc. B* 367:2594-606, 2012; Studer *et al.*, *Science* 354:229-32, 2016). Finally, the existence of dispensable interactions is well-justified by standard molecular evolution theory. Dispensable interactions are introduced into and maintained in the interactome by non-adaptive processes, when purifying selection is not strong enough to maintain the perfect interactome in the presence of mutation and genetic drift (Landry *et al.*, *Trends Genet.* 25:193-7, 2009).

We completely agree with the reviewer that the deep mutational scanning papers by the Bolon and Fraser labs demonstrate that fitness measurements under laboratory conditions do not accurately reflect natural selective pressures in changing environments over evolutionary timescales. Hence, instead of using fitness measurements under laboratory conditions, we use

population genetic datasets to accurately measure selective pressures and fitness effects of mutations. In our study, mutations known to cause Mendelian diseases are considered deleterious, whereas common mutations carried by healthy individuals are considered neutral.

We completely agree with the reviewer on the importance of macromolecular crowding in modulating interactions. In this study, we make the reasonable assumption that macromolecular crowding exerts similar thermodynamic effects on each binary protein-protein interaction before and after mutation. Crowding effects can be modulated by several factors, including protein shape (Guseman *et al.*, *PNAS* 115:10965-70, 2018). While beyond the scope of this study, it will be fascinating for future work to systematically investigate the effect of protein shape on crowding at the interactome scale.

We have revised our manuscript accordingly to include the relevant literature citations and discuss in detail the biological meaning of dispensable interactions, the limitations of fitness measurements under laboratory conditions, and the importance of macromolecular crowding in modulating interactions (Pages 2-3 Lines 47-52; Pages 19-20 Lines 361-378).